# Dual-Responsive Supramolecular Chiral Assemblies from Amphiphilic Dendronized Tetraphenylethylenes

**DOI:** 10.3390/molecules28186580

**Published:** 2023-09-12

**Authors:** Jianan Zhang, Xueting Lu, Wen Li, Afang Zhang

**Affiliations:** International Joint Laboratory of Biomimetic and Smart Polymers, School of Materials Science and Engineering, Shanghai University, Nanchen Street 333, Shanghai 200444, China

**Keywords:** supramolecular chirality, dendronization, self-assembly, thermoresponsive behavior, aggregation-induced emission

## Abstract

Supramolecular assembly of amphiphilic molecules in aqueous solutions to form stimuli-responsive entities is attractive for developing intelligent supramolecular materials for bioapplications. Here we report on the supramolecular chiral assembly of amphiphilic dendronized tetraphenylethylenes (TPEs) in aqueous solutions. Hydrophobic TPE moieties were connected to the hydrophilic three-fold dendritic oligoethylene glycols (OEGs) through a tripeptide proline–hydroxyproline–glycol (POG) to afford the characteristic topological structural effects of dendritic OEGs and the peptide linker. Both ethoxyl- and methoxyl-terminated dendritic OEGs were used to modulate the overall hydrophilicity of the dendronized TPEs. Their supramolecular aggregates exhibited thermoresponsive behavior that originated from the dehydration and collapse of the dendritic OEGs, and their cloud point temperatures (*T*_cp_s) were tailored by solution pH conditions. Furthermore, aggregation-induced fluorescent emission (AIE) from TPE moieties was used as an indicator to follow the assembly, which was reversibly tuned by temperature variation at different pH conditions. Supramolecular assemblies from these dendronized amphiphiles exhibited enhanced supramolecular chirality, which was dominated mainly by the interaction balance between TPE with dendritic OEG and TPE with POG moieties and was modulated through different solvation by changing solution temperature or pH conditions. More interestingly, ethoxyl-terminated dendritic OEG provided a much stronger shielding effect than its methoxyl-terminated counterpart to prevent amino groups within the peptide from protonation, even in strong acidic conditions, resulting in different responsive behavior to the solution temperature and pH conditions for these supramolecular aggregates.

## 1. Introduction

Supramolecular assembly has been proven to be an important route for obtaining higher-ordered structures through molecular design [1,2,3], which has also been demonstrated to be a useful route for fabricating supramolecular materials [4,5,6]. The driving forces that govern the self-assembly of amphiphiles arise from at least three major energy contributions: hydrophobic interactions [7,8,9], hydrogen bonding [10], and repulsions between the segments [11,12]. The final assemblies, including nanotubes [13,14], fibers [15,16], and bands [17,18] reflect a delicate balance of each of these energy contributions [19,20]. The effects of topological structures on the supramolecular assembly are interesting due to the varied interaction balance between various structure moieties through different packing fashions.

Controlling the self-assembly of amphiphilic molecules by altering the local environment has been proven to be useful in manipulating supramolecular processes and supramolecular structures. One way to realize this is by using external stimuli to trigger off or alter the amphiphilicity of a molecule, resulting in varied assembly behavior and forming aggregates with different morphologies. Therefore, stimuli-responsive supramolecular assembly is receiving considerable attention and has been found promising in fabricating intelligent supramolecular materials [21,22,23]. Among them, solvent polarity-responsive [24,25], photo-responsive [26,27,28], thermoresponsive [29,30], and pH-responsive supramolecular assemblies [31,32] have been mostly investigated. The combination of chirality with stimuli-responsive supramolecular assembly affords a convenient way to modulate the supramolecular chirality and simultaneously mediate the supramolecular assembly [33,34,35]. As a clean tool, temperature has often been used as a stimulus in supramolecular chiral assembly to fabricate thermoresponsive chiral materials [36]. Furthermore, supramolecular chiral assembly in aqueous solutions is attractive for bioapplications [37,38], chiral sensing and separation [39,40], and chiral templates and optics [41,42]. However, manipulating the supramolecular chiral assembly in water remains a challenge, especially when multiple responses are targeted.

Tetraphenylethylenes (TPEs) are a typical class of aggregation-induced luminescence (AIE) molecules [43,44]. The combination of TPEs to construct amphiphilic molecules will endorse excellent fluorescence properties [45,46,47], which can even be used as molecular signaling to “visualize” the assembly through the AIE effect [48,49]. Recently, we reported supramolecular chiral assembly of dendronized TPEs carrying the dipeptide alanine–glycine in aqueous solutions and found that solvation through varying water/THF ratios dominated the chiral assembly, resulting in supramolecular chiral spheres with tunable AIE and compositions [50]. Surprisingly, increasing the water ratio induced the dendritic components to shift from the interior of the aggregates onto the peripherals, resulting in higher phase transition temperatures for the aggregates, opening a novel protocol to manipulate the assembly and simultaneously the supramolecular morphologies. This also facilitates a way to deliver the loaded molecules or moieties through supramolecular assembly in a controlled fashion, simply mediated by temperature. As a continuation, in the present report, TPE is used as the hydrophobic moiety for the construction of a novel class of dendronized amphiphiles to trigger supramolecular assembly in water. Three-fold dendritic oligoethylene glycols (OEGs) with either ethoxyl or methoxyl terminals are used as the hydrophilic moieties to afford dendronized amphiphiles with characteristic topological features and different overall hydrophilicity. Crowded OEG chains may endow the aggregates with thermoresponsiveness [51,52,53], creating a class of thermoresponsive and fluorescence supramolecular chiral assemblies. Instead of peptides from alanine and glycine, which were previously used, here proline–hydroxyproline–glycine (POG) is used as the chiral source for inducing supramolecular chirality, which is the most abundant peptide sequence in collagen. The hydroxy group in POG provides a chance to attach the dendritic OEGs to the side instead of the end of the tripeptide, creating a chance to examine the topological effects of the amphiphilic molecules on their supramolecular chiral assemblies. Furthermore, the tripeptide POG has the amino terminal free, which provides a chance to modulate the overall hydrophilicity of the dendronized TPEs through solution pH conditions.

## 2. Results and Discussion

### 2.1. Synthesis and Characterization

OEG-based dendronized TPEs carrying three-fold dendritic OEGs with two different terminals (methoxyl- or ethoxyl-) were designed, aiming at (1) supramolecular assembly in the aqueous phase, (2) varied hydrophilicity to modulate the phase transition temperatures, and (3) supramolecular chirality enhancement through thermoresponsiveness. Furthermore, thermally induced phase transitions may provide an additional tool to investigate crowding effects on the supramolecular assembly of these thermoresponsive amphiphiles. Here, tripeptide POG is used as a chirality source, which is the major component in collagens, and its amino terminal is used as a pH regulator, affording the targeted molecules with pH responsiveness. TPE with aggregation-induced emission characteristics is used as the hydrophobic moiety. Different from the conventional linear arrangement for different segments in constructing the amphiphilic molecules, here we design to have both hydrophilic dendritic OEG and hydrophobic TPE segments “face-to-face” arranged beside POG, aiming at different tendencies for interactions between these two segments while at the same time providing less steric hindrance for the free amino group to exhibit enhanced pH responsiveness. The synthesis of these dendronized amphiphiles is illustrated in Figure 1. Starting from the tripeptide **Boc-POG-OMe**, mesylation gave the known intermediate **Boc-PO(Ms)G-OMe**, which was converted into the azide **Boc-PO(N_3_)G-OMe**. This compound was hydrolyzed by LiOH to transform from the ester into the corresponding acid **Boc-PO(N_3_)G-OH**. This acid was transferred into **Boc-PO(N_3_)G-TPE** through amidation with 1-(4-aminophenyl)-1,2,2-triphenylethene (TPE-NH_2_) in the presence of DiPEA as a base and EDC·HCl/HOBt as coupling agents. Dendronized amphiphiles were achieved via a Cu(I)-catalyzed azide–alkyne “click reaction” from **Boc-PO(N_3_)G-TPE** with the dendritic OEGs **MeG1** or **EtG1** in the presence of L-ascorbic acid sodium salt as catalyst. The azide–alkyne “click reaction” was performed in highly concentrated solutions of t-BuOH/H_2_O, which are good solvents for the peptides. After Boc- deprotection with TFA, dendronized TPEs with free amine in the peptide POG were obtained. All new compounds were characterized by ^1^H and ^13^C NMR spectroscopy (Appendix A), as well as high-resolution mass spectrometry (Appendix A), to confirm their structures.

### 2.2. Supramolecular Aggregations in Aqueous Solutions

Fluorescence (FL) spectroscopy was first used to follow the supramolecular aggregation of these dendronized TPEs in aqueous solutions at different pH conditions. As shown in Figure 1, significant AIEs corresponding to TPE moieties were observed from both **H-PO(Me)G-TPE** and **H-PO(Et)G-TPE**, indicating intensive aggregation of these molecules in aqueous solutions with TPE packed in a tight fashion. However, the emission transitions for the two dendronized TPEs according to pH variation are surprisingly different. For **H-PO(Me)G-TPE** carrying a more hydrophobic dendritic OEG pendant, the emission intensity at 480 nm increased significantly with an increase in solution pH, indicating a higher pH condition (less protonated amino groups) is favorable to enhance the aggregation of TPE moieties from **H-PO(Me)G-TPE**. While for **H-PO(Et)G-TPE**, which carries a more hydrophobic dendritic OEG pendant, the emission intensity at 480 nm decreased obviously with an increase in solution pH, suggesting that de-aggregation happened or at least the packing of TPE units became less compact when solution pH increased. This difference between **H-PO(Me)G-TPE** and **H-PO(Et)G-TPE** is interesting. For the hydrophilic **H-PO(Me)G-TPE**, it tends to be dissolved in water preferentially. Only when protonation of the amine from POG was reduced with an increase in solution pH conditions did its hydrophobicity increase accordingly, which induced intensive assembly for the enhanced AIE effect. While for the more hydrophobic **H-PO(Et)G-TPE**, it can assemble well in water without help from the protonation from the amino group. Protonation of the amino group from POG under strong acidic conditions should have enhanced its amphiphilicity to assemble in water, leading to an extensive AIE effect. This can be changed through an increase in solution pH to reduce the protonation of the amino group and make **H-PO(Et)G-TPE** too hydrophobic. The high hydrophobicity of **H-PO(Et)G-TPE** makes its assembly in water pack in less ordered structures, resulting in weakened AIE. AFM measurements reveal that supramolecular assembly of **H-PO(Me)G-TPE** formed spheres with sizes in the range of a few hundreds of nanometers at pH 5 (Figure 1c), but long fibers with diameters in the range of 25–35 nm were observed from its solution at pH 7 (Figure 1d). This morphology difference should originate from the amphiphilicity difference of the dendronized TPE at different pH conditions, which also indicates that the assembly morphologies of the dendronized amphiphiles can be simply controlled through solution pH conditions.

Solution pH variation is actually equivalent to the solvation changes in these dendronized TPEs. The amino unit in tripeptide POG is the only structural unit sensitive to solution pH variation, which should be protonated differently at different pH conditions and, consequently, contribute varied hydrophilicity to the amphiphilic dendronized TPEs. This hydrophilicity variation should have changed their solvation in water, mediating their different aggregation propensities according to solution pH conditions. Therefore, different aggregation behaviors for **H-PO(Me)G-TPE** and **H-PO(Et)G-TPE** should be due to different protonation tendencies of the amino unit from POG. As we reported previously, dendritic OEGs can shield protonation of encapsulated moieties through crowding effects and act as a molecular envelop for protection of the guest segments or guest molecules in aqueous solutions [54,55,56,57]. In the present case, dendritic OEG was covalently linked to the tripeptide, which should have provided a shielding effect for protonation of the amino group in the vicinity, leading to a different propensity for the amine from the tripeptide to be protonated at different pH conditions.

To track the aggregation on the molecular level with variation in solvation through pH changes, ^1^H NMR spectra of **H-PO(Me)G-TPE** and **H-PO(Et)G-TPE** in aqueous solutions at different pH were recorded, and the results are shown in Figure 2. Proton signals from **H-PO(Me)G**-**TPE** were well resolved in acidic conditions at pH 4, indicating weak aggregation. However, proton signals from TPE and OEG moieties became poorly resolved and significantly broader with an increase in solution pH and eventually immersed in the baseline at pH 7, indicating enhanced aggregation with an increase in solution pH. However, proton signals from **H-PO(Et)G-TPE** remained well-resolved in the range of pH 1–7, suggesting solvation of this molecule is not so relevant to solution pH conditions. Actually, proton signals slightly increased their resolution with an increase in solution pH from 1 to 6. That is why the AIE fluorescence intensities decreased with an increase in solution pH for **H-PO(Et)G-TPE**. Above, it is indicated that the ethoxy-terminated dendritic OEG motif provides much better protonation shielding ability than its methoxy counterpart, the amino group from POG.

To examine the interaction between TPE and OEG moieties in aqueous solutions, NOESY spectra of **H-PO(Me)G-TPE** and **H-PO(Et)G-TPE** in D_2_O were recorded. As shown in Appendix A, obvious cross-couplings between TPE and OEG moieties were observed at acidic conditions for both dendronized TPEs, indicating intensive interactions between TPE and OEG units. The signal intensities from cross-coupling between TPE and OEG moieties for **H-PO(Me)G-TPE** became weak with an increase in solution pH, as shown in Appendix Ab,d, which should be due to the intensive aggregation of the whole molecule, leading to reduced proton signal intensities. Differently, as shown in Appendix A for **H-PO(Et)G-TPE**, no significant change in cross coupling between TPE and OEG moieties was observed with an increase in solution pH from 1 to 6, which further proves that more hydrophobic ethoxy-terminated dendritic OEGs exhibit better ability than the more hydrophilic methoxy-terminated dendritic OEG moieties in shielding the protonation of the amino group from POG.

### 2.3. Thermoresponsive Properties of the Assemblies

One characteristic feature of dendritic OEGs is their ability to afford dendronized polymers with unprecedented thermoresponsive properties [40,41,42,43,44]. Therefore, we expect that supramolecular assemblies from the dendronized TPEs may also inherit this thermoresponsive feature. Actually, responsive or smart self-assembly is a common process for biomacromolecules to generate biofunctions or conduct bioactivities. Therefore, UV/vis spectroscopy was utilized first to examine the thermoresponsive properties of **H-PO(Me)G-TPE** and **H-PO(Et)G-TPE** assemblies in aqueous solutions at different pH, and the transmittance changes at 700 nm with temperature are shown in Figure 3. With an increase in temperature, the transparent solutions became turbid without precipitation and returned to clear solutions again when the temperature was decreased to room temperature. The transition temperature for the aqueous solutions from clear to turbid is defined as the cloud point temperature (*T*_cp_), which was found to be dependent on the hydrophilicity of dendritic OEGs and also related to solution pH conditions. For **H-PO(Me)G-TPE** with more hydrophilic methoxy terminals, the *T*_cp_ from its aggregates was found to be 83.0 °C at pH 4, which greatly decreased to 65.0 °C at pH 5, 61.0 °C at pH 6, and 46.0 °C at pH 7. This indicates that more protonated amine from POG at more acidic conditions affords the molecule much higher hydrophilicity. However, for **H-PO(Et)G-TPE** with more hydrophobic ethoxy terminals, *T*_cp_s for the assemblies at pH 1, 2, 3, 4, 5, and 6 all remained at 28 °C. This indicates that the protonation difference of the amine in different pH conditions can be negligible in **H-PO(Et)G-TPE**. The different shielding effect on protonation of the amine between **H-PO(Me)G-TPE** and **H-PO(Et)G-TPE** is amazing, which suggests more hydrophobic ethoxyl-terminated dendritic OEGs exhibit more intensive interactions with the peptide moiety to prevent it from hydration.

Thermally induced collapse and aggregation may have an impact on the packing of TPE units within the aggregates, mediating different AIE effects. Therefore, FL spectra of aggregates from **H-PO(Me)G-TPE** and **H-PO(Et)G-TPE** at different temperatures were recorded at different pH. As shown in Appendix A for the FL spectra and Figure 3c,d for the plots of the FL intensities against temperature, the intensity of AIE around 480 nm from both dendronized TPEs declined with increasing solution temperature, and the emission was nearly quenched at elevated temperatures above their phase transition temperatures. This thermally induced AIE quenching is reversible, and the emission can be recovered immediately after cooling down. Before the *T*_cp_, increasing temperature leads to higher molecular mobility, which reduces the AIE effect. With an increase in solution temperature for the *T*_cp_, collapsed dendritic OEGs formed a more hydrophobic domain, which strengthened interaction with the hydrophobic TPE moieties to prevent them from compact packing, resulting in reduced AIE effects. Higher above the *T*_cp_, more dehydrated and collapsed dendritic OEGs interfered more intensively with TPE moieties to prevent them from densely packing, resulting in quenching of the AIE effects. On the other side, TPE moieties became more mobile due to the high temperature, which also contributed to the disordering of TPEs in the collapsed matrix and reduced AIE effects. Since the collapse-enhanced interactions between dendritic OEGs and TPE moieties can be switched on and off through temperature changes across the phase transition point, intensive AIE emission can be restored once the solution temperature is reduced back to room temperature. Therefore, dominant interactions among the TPE moieties enhance the AIE emission, while dominant interactions between dendritic OEG and TPE moieties cause the AIE quenching.

^1^H NMR spectra at different temperatures were recorded to examine at the molecular level the aggregations of these dendronized TPEs. As shown in Appendix A for **H-PO(Me)G-TPE**, proton signals were very broad at low temperatures, indicating its extensive aggregation. With the increase in solution temperature, broad proton signals corresponding to both dendritic OEG and TPE all became better resolved due to the thermally enhanced mobility of the molecule. With an increase in solution temperature above its *T*_cp_, proton signals from dendritic OEG split into two groups, one broad and another even better resolved. This can be clearly seen from the proton signals corresponding to terminal methyl groups in dendritic OEG. The broad signals came from the thermally induced collapse of the OEGs, while the resolved signals came from the well-soluble ones. The above observation suggests that dendritic OEGs play two roles during the phase transition process: one to decorate the aggregates and assist their dissolution in water from precipitation, and another to be involved in co-aggregation with TPE moieties to reduce the surface tension for the TPE domain within the aqueous phase. A similar phenomenon was observed for **H-PO(Et)G-TPE**, as shown from its ^1^H NMR spectra at different temperatures (Appendix A).

NOESY spectra of **H-PO(Me)G-TPE** and **H-PO(Et)G-TPE** were recorded to track the assembly on the molecular level across the thermally induced phase transitions. As shown in Appendix A for **H-PO(Me)G-TPE**, significant cross-coupling was observed between TPE and dendritic OEG moieties with increasing temperature. In this condition, collapsed OEGs formed more hydrophobic domains to strengthen interactions with the hydrophobic TPEs, which led to a disordered arrangement of the TPEs in the matrix, resulting in AIE quenching. Differently, as shown in Appendix A for **H-PO(Et)G-TPE**, a gradual weakened cross-coupling was observed between TPE and dendritic OEG moieties with an increase in temperature. This suggests that, at low temperatures, swollen dendritic OEG was more involved in co-aggregation with TPE moieties, but dehydrated dendritic OEG tends to decorate the aggregates and assist their dissolution in water at elevated temperatures.

### 2.4. Supramolecular Chirality

Chiroptical properties of the supramolecular assemblies from the dendronized TPEs were investigated by circular dichroism (CD) spectroscopy. As shown in Figure 4, induced Cotton effects in the range of 292 nm corresponding to TPE chromophore were observed for aggregates from both **H-PO(Me)G-TPE** and **H-PO(Et)G-TPE** in aqueous solutions. The signal intensities of the Cotton effects increased greatly with solution pH. Since induced chirality from TPE moieties should be dependent on how strong the interaction between non-chiral TPE and the chiral POG moieties is, the above results indicate that a neutral condition is favorable to enhance the interaction between TPE and less protonated POG moieties to induce the ordered and compact packing of TPE moieties. Combining the results of the pH-dependent AIE of these dendronized TPEs discussed previously, we propose that enhanced aggregation due to less protonation of the amine within the POG peptide at neutral conditions is supportive of tightly packing TPE units in the hydrophobic matrix and simultaneously strengthens the interaction of the peptide segment with TPE moieties for efficient chiral induction. It is worthwhile to point out that, by comparing Figure 4a with Figure 4b, induced chirality from the assembly of **H-PO(Me)G-TPE** is much stronger than that from **H-PO(Et)G-TPE**, indicating enhanced amphiphilicity due to the larger hydrophilicity difference between methoxyl-terminated dendritic OEGs and the hydrophobic TPE, which is helpful for tight and ordered packing of TPE moieties in the aggregates.

Chiroptical properties of the supramolecular assemblies at different temperatures were examined by CD spectroscopy. As shown in Appendix A and Figure 4c for **H-PO(Me)G-TPE**, Cotton effect from TPE moieties decreased gradually with an increase in solution temperature and declined sharply at slightly higher temperatures. Surprisingly, the aggregates became chirality silent before their *T*_cp_. This is interesting and suggests that supramolecular chirality in the assemblies was very sensitive to temperature, even below its *T*_cp_ before the dendritic OEG started to dehydrate and collapse. With an increase in solution pH, supramolecular chirality started to diminish at a slightly higher temperature point, as shown in Figure 4c. However, at neutral conditions, increasing temperature reduced Cotton effect at a much lower temperature, indicating that the assembly of **H-PO(Me)G-TPE** in a more hydrophobic state is more sensitive to temperature. This temperature-dependent chirality transition is reversible since a decrease in temperature to room temperature can recover the supramolecular chirality. This sensitivity of supramolecular chirality to temperature is even more pronounced for **H-PO(Et)G-TPE**. As shown in Appendix A and Figure 4d, Cotton effect of the aggregates from **H-PO(Et)G-TPE** sharply declined and became chirality silent at temperatures much below its *T*_cp_. Above indicates that the supramolecular chirality of the aggregates from these dendronized TPEs is not relevant to their thermal phase transitions, i.e., independent of the dehydration and collapse of the dendritic OEGs. Instead, the supramolecular chirality should be mainly related to the interaction balance between TPE with dendritic OEG and TPE with POG moieties. POG can be wrapped easily by the dendritic OEG moieties due to the arrangement of different units on the dendronized TPEs. Since the OEG unit was connected to the side, not the end of the tripeptide, this makes it much easier for the OEG units to strongly interact and wrap with the tripeptide POG, which should have rendered the interaction between TPE and the tripeptide much weaker and more sensitive to the external environment, including temperature changes or solution pH conditions. This finding reveals that the topological structures of amphiphilic molecules play an important role in mediating their chiral assembly.

## 3. Conclusions

Manipulating the amphiphilicity of a molecule through external stimuli is important in modulating its supramolecular assembly, which may pave a novel route to developing intelligent supramolecular materials for various applications. Assisted by fluorescence, UV/vis, NMR, and AFM spectroscopies, supramolecular assemblies of amphiphilic dendronized TPEs in aqueous solutions were investigated. These dendronized TPEs carried either methoxyl- or ethoxyl-terminated three-fold dendritic OEG through a POG tripeptide. Hydrophobic TPE moieties initiated the aggregation in the poor solvent water, and AIE and induced chirality from TPE were used to follow the assembly. Dendritic OEGs provided a complementary driving force to interact with TPE or the peptide moieties and therefore played a critical role in modulating the assembly. Crowded OEG moieties acted as a “skin” to form a supramolecular dendritic shell to encapsulate the guest TPE moieties, affording the aggregates characteristic thermoresponsiveness. More interestingly, methoxyl- and ethoxyl-terminated dendritic OEG provided a distinct shielding effect for protonation of the amine within POG, resulting in different pH-responsiveness for the aggregates from these dendronized TPEs. Phase transition temperature of **H-PO(Me)G-TPE** increased greatly with decrease in solution pH due to enhanced protonation of the amine from POG but remained constant for **H-PO(Et)G-TPE** at pH from 7 to 1, indicating that strong interactions between ethoxy-terminated OEGs and the POG peptide have prevented the amine from protonation even in strong acidic conditions. For **H-PO(Me)G-TPE**, whose assembly in water is highly dependent on solution pH conditions, chiral spheres were observed at pH 7, but long fibers were achieved at pH 7, indicating easy tunability of the assemblies through solution pH conditions. The principles developed in the present work have not only provided a general methodology for the fabrication of intelligent supramolecular chiral aggregates in the aqueous phase through dendronization with three-fold dendritic OEGs but also demonstrated the importance of topological structures in the supramolecular assembly, which are important for mimicking biological functions and may find promising applications in intelligent chiral materials, such as chiral recognitions and chiral catalysis.

## 4. Materials and Methods

### 4.1. Materials

**Boc-POG-OMe** [58], **MeG1** [59], and **EtG1** [59] were synthesized according to our previous reports. Tosyl chloride (MsCl), EDC·HCl, NaSAC, and TPE-NH_2_ were purchased from TCI (Tokyo, Japan). Dry DMF, DIPEA, copper sulfate pentahydrate, TFA, and DMAP were purchased from Acros. Pyridine, lithium hydroxide monohydrate, t-BuOH, and methanol were purchased from China National Pharmaceutical Group Corporation. HOBt was purchased from GLS. DCM was distilled from CaH_2_ for drying. All reactions were run under a nitrogen atmosphere. Other reagents and solvents were of reagent grade and used without further purification. Macherey–Nagel precoated TLC plates (silica gel 60 G/UV254, 0.25 mm) were used for the thin-layer chromatography (TLC) analysis. Silica gel 60 M (Macherey-Nagel, Düren, Germany, 0.040–0.063 mm, 200–300 mesh) was used as the stationary phase for column chromatography.

### 4.2. Instrumentation and Measurements

^1^H and ^13^C NMR spectra were recorded on a Bruker AV 500 (^1^H: 500 MHz; ^13^C: 125 MHz) spectrometer. CD measurements were performed on a JASCO J-815 spectropolarimeter (Tokyo, Japan) with a thermos-controlled 1 mm quartz cell (three accumulations, “continuous” scanning mode, scanning speed: 200 nm·min^−1^ data pitch: 0.2 nm; response: 1 s; bandwidth: 2.0 nm). Turbidity measurements were carried out on a PE UV-vis spectrophotometer (Lambda 35. Norwalk, CT, USA) equipped with a thermo-controlled bath. Polymer aqueous solutions were placed in the spectrophotometer (path length 1 cm) and heated or cooled at a rate of 1.0 °C·min^−1^. Absorptions of the solution at λ = 700 nm were recorded per 5 s. The cloud point temperature (*T*_cp_) was determined as the one at which the transmittance at λ = 700 nm had reached 50% of its initial value. The pH value was measured with a Mettler-Toledo Seven Compact220 pH meter and an InLab Flex-Micro semi-microelectrode (Hong Kong, China. After 4-point calibrations at pH 10.00, 7.00, 4.01, and 2.00).

### 4.3. Synthesis

*General Procedure for Mesylation* (a): The respective hydroxy compound (2.50 mmol) was dissolved in dry pyridine (5 mL) and cooled in an ice bath. MsCl (10.00 mmol) was added in one portion, and the reaction mixture was stirred for 4 h in the ice bath and then for 6 h at room temperature. The reaction was then quenched by the addition of methanol (2 mL). Evaporation of the solvent gave a residue, which was dissolved in ethyl acetate (30 mL). The organic phase was washed successively with NaHCO_3_ (1 M), citric acid (1 M), and brine. The mixture was dried over MgSO_4_, filtered, and the solvent removed. Purification of the residue by column chromatography with DCM/methanol (60:1, *v*/*v*) afforded the mesylated compound as light-yellow or colorless needles. *General Procedure for Azide Substitution from the Mesylated Compound* (b): The mesylated compound (2.02 mmol) and NaN_3_ (7.13 mmol) were stirred in dry DMF (5 mL) at 45–55 °C overnight. The solvent was evaporated, and the residue was taken up with DCM (25 mL) and H_2_O (20 mL). The organic phase was washed with H_2_O until neutral, and then successively with NH_4_Cl (1 M) and brine. The mixture was dried over MgSO_4_, filtered, and the solvent removed. Purification of the residue by column chromatography with DCM/methanol (80/1, *v*/*v*) afforded the azide as a colorless oil. *General Procedure for Saponification of Methyl Ester by LiOH* (c): LiOH·H_2_O (12.30 mmol) was added to a solution of methyl ester (0.82 mmol) in methanol (10 mL) and water (2 mL) at −5 °C with stirring. The reaction temperature was then allowed to rise to room temperature. After the mixture was stirred for 4 h, the solvents were evaporated in vacuo at room temperature, and the residue was dissolved in DCM. The pH of the solution was adjusted carefully to pH 2–3 with 10% KHSO_4_. The organic phase was washed with brine. All the aqueous phases were extracted with DCM three times. The combined organic phase was dried over MgSO_4_. After filtration, the solvent was evaporated in vacuo. Purification of the residue by column chromatography with dichloromethane/methanol (30:1, *v*/*v*) afforded the corresponding acid as colorless crystals. *General Procedure for Amidation* (d): The acid compounds (0.24 mmol), HOBt (0.36 g, 0.26 mmol), TPE-NH_2_ (0.10 g, 0.29 mmol), and DIPEA (0.62 g, 0.48 mmol) were dissolved in dry DCM (16 mL) at 0 °C, and the solution was stirred for 20 min before addition of EDC·HCl (0.92 g, 0.48 mmol). The resulting mixture was stirred for 12 h at room temperature. After being washed successively with saturated solutions of NaHCO_3_ and 10% KHSO_4_, the organic phase was dried over magnesium sulfate. After evaporation under reduced pressure of the organic solvent, the crude product was purified by column chromatography with DCM/methanol (100:1, *v*/*v*) to yield the targeted compound as a colorless oil. *General Procedure for “click reaction”* (e): Azide (0.21 mmol) and **MeG1** or **EtG1** (0.18 mmol) were dissolved in t-BuOH/H_2_O (*v*/*v* = 1:1), and the solution was stirred for 20 min before addition of NaSAC (0.16 g, 0.08 mmol) and CuSO_4_·5H_2_O (0.05 g, 0.02 mmol). The resulting mixture was stirred for 12 h at room temperature. The solvents were evaporated in vacuo at room temperature, and the residue was dissolved in ethyl acetate. The organic phase was washed with brine. The combined organic phase was dried over MgSO_4_. After filtration, the solvent was evaporated in vacuo. Purification of the residue by column chromatography with DCM/methanol (50:1, *v*/*v*) afforded the corresponding acid as colorless crystals. *General Procedure for Boc Removal with TFA* (f): TFA (4.48 mmol) was added to a solution of Boc-protected compound (0.11 mmol) in DCM (5 mL) at 0 °C, and the mixture was stirred for 6 h. Then, an excess amount of methanol was added to quench the reaction. Evaporation of the solvents in vacuo yielded the deprotected product as colorless, needlelike crystals. *Tert-butyl (S)-2-((2S,4R)-2-((2-methoxy-2-oxoethyl)carbamoyl)-4-((methylsulfonyl) oxy) pyrrolidine-1-carbonyl)pyrrolidine-1-carboxylate* (**Boc-PO(Ms)G-OMe**). According to general procedure, *a* from MsCl (1.14 g, 10.00 mmol), DMAP (0.24 g, 2.00 mmol), and **Boc-POG-OMe** (1.00 g, 2.50 mmol) in dry pyridine (5 mL), the compound **Boc-PO(Ms)G-OMe** was afforded as a colorless product in a nearly quantitative yield (1.02 g, 85%). ^1^H NMR (DMSO-*d*_6_): δ = 1.25–1.38 (2s, 9H, H-Boc), 1.75 (m, 2H, CH_2_), 2.13 (m, 2H, CH_2_), 2.41 (m, 2H, CH_2_), 3.23 (m, 2H, CH_2_), 3.28 (m, 3H, CH_3_), 3.62 (t, 3H, CH_3_), 3.75–3.80 (m, 2H, CH_2_), 3.89–4.01 (m, 2H, CH_2_), 4.40–4.48 (m, 2H, CH), 5.36 (s, 1H, CH), 8.41–8.44 (m, 1H, NH). ^13^C NMR (DMSO-*d*_6_): δ = 23.06, 23.70, 28.13, 28.27, 28.49, 29.30, 30.87, 35.52, 36.05, 37.73, 37.90, 39.60, 40.62, 46.37, 46.55, 51.81, 52.54, 52.70, 57.62, 57.65, 57.69, 57.72, 78.44, 78.59, 80.09, 153.20, 153.51, 162.40, 170.25, 170.38, 170.74, 171.38. HR-MS (ESI): *m*/*z* calcd for C_19_H_31_N_3_O_9_NaS [M + Na]^+^: 500.1673, found: 500.1672. *Tert-butyl (S)-2-((2S,4S)-4-azido-2-((2-methoxy-2-oxoethyl)carbamoyl)pyrrolidine-1-carbonyl)pyrrolidine-1-carboxylate* (**Boc-PO(N_3_)G-OMe**). According to general procedure, *b* from **Boc-PO(Ms)G-OMe** (0.96 g, 2.02 mmol) and NaN_3_ (0.46 g, 7.13 mmol) in dry DMF (5 mL), compound **Boc-PO(N_3_)G-OMe** was afforded as a colorless product in a nearly quantitative yield (0.53 g, 60%). ^1^H NMR (DMSO-*d*_6_): δ = 1.28 and 1.37 (2d, 9H, H-Boc), 1.69–1.93 (m, 4H, CH_2_), 2.07–2.48 (m, 2H, CH_2_), 3.25–3.38 (m, 2H, CH_2_), 3.58, 3.62 (2s, 3H, CH_3_), 3.77–3.83 (m, 2H, CH_2_), 3.83–3.88 (m, 2H, CH_2_), 3.89–3.91 (m, 2H, CH_2_), 3.94–3.99 (m, 2H, CH_2_), 4.05–4.08 (m, 2H, CH_2_), 4.32–4.43 (m, 3H, CH), 8.16–8.21 (m, H, NH). ^13^C NMR (DMSO-*d*_6_): δ = 23.13, 23.76, 24.21, 28.05, 28.11, 28.22, 28.49, 28.65, 29.93, 33.76, 35.46, 36.05, 37.73, 37.78, 37.90, 39.10, 39.27, 39.43, 39.60, 39.77, 39.93, 40.10, 40.18, 40.62, 42.62, 46.37, 46.46, 46.55, 51.81, 52.47, 52.54, 52.70, 56.56, 57.37, 57.62, 57.65, 57.69, 57.72, 57.81, 60.30, 78.34, 78.40, 78.44, 78.79, 80.09, 106.71, 153.20, 153.40, 153.45, 153.46, 153.51, 154.16, 162.40, 169.76, 169.90, 170.04, 170.22, 170.25, 170.38, 170.74, 170.76, 170.93, 171.06, 171.08, 171.18, 171.35. HR-MS (ESI): *m*/*z* calcd for C_18_H_29_N_6_O_6_ [M + H]^+^: 425.2143, found: 425.2140. *((2S,4S)-4-azido-1-((tert-butoxycarbonyl)-L-prolyl)pyrrolidine-2-carbonyl)glycine* (**Boc-PO(N_3_)G-OH**). According to general procedure *c* from LiOH·H_2_O (0.50 g, 12.30 mmol) and **Boc-PO(N_3_)G-OMe** (0.35 g, 0.82 mmol) in methanol (10 mL) and water (2 mL), compound **Boc-PO(N_3_)G-OH** was afforded as a colorless product in a nearly quantitative yield (0.27 g, 80%). ^1^H NMR (DMSO-*d*_6_) δ = 1.30 and 1.37 (2s, 9H, H-Boc), 1.72–1.96 (m, 4H, CH_2_), 2.09–2.46 (m, 2H, CH_2_), 3.25–3.30 (m, 2H, CH_2_), 3.64–3.72 (m, 2H, CH_2_), 3.77–3.82 (m, 2H, CH_2_), 3.94–3.98 (m, 2H, CH_2_), 4.04–4.08 (m, 2H, CH_2_), 4.36–4.43 (m, 3H, CH), 8.00–8.05 (m, 1H, NH), 12.59 (s, 1H, OH). ^13^C NMR (DMSO-*d*_6_): δ = 23.17, 23.79, 24.27, 28.08, 28.26, 28.70, 29.11, 29.14, 29.65, 33.76, 35.47, 39.60, 40.81, 46.49, 46.59, 47.00, 51.15, 51.25, 52.08, 55.02, 57.43, 57.63, 58.14, 58.67, 78.46, 78.62, 79.42, 153.07, 153.54, 154.18, 170.74, 170.78, 170.90, 171.19, 171.40. HR-MS (ESI): *m*/*z* calcd for C_17_H_26_N_6_O_6_Na [M + Na]^+^: 433.1806, found: 433.1809. *Tert-butyl (S)-2-((2S,4S)-4-azido-2-((2-oxo-2-((4-(1,2,2-triphenylvinyl)phenyl)amino) ethyl)carbamoyl)pyrrolidine-1-carbonyl)pyrrolidine-1-carboxylate* (**Boc-PO(N_3_)G-TPE**). According to general procedure *d* from **Boc-PO(N_3_)G-OH** (0.10 g, 0.24 mmol), TPE-NH_2_ (0.10 g, 0.29 mmol), HOBt (0.36 g, 0.26 mmol), DIPEA (0.62 g, 0.48 mmol), EDC·HCl (0.92 g, 0.48 mmol) in dry DCM (16 mL), compound **Boc-PO(N_3_)G-TPE** was afforded as a colorless product in nearly quantitative yield (0.16 g, 89%). ^1^H NMR (DMSO-*d*_6_): δ = 1.31 (t, 9H, H-Boc), 1.44–1.89 (m, 4H, CH_2_), 2.09–2.20 (m, 2H, CH_2_), 2.95–3.37 (m, 2H, CH_2_), 3.61–3.75 (m, 2H, CH_2_), 3.88–4.13 (m, 2H, CH_2_), 4.29–4.46 (m, 3H, CH), 6.87–7.64 (m, 19H, CH), 8.53, 8.75 (2t, 1H, NH), 9.31, 9.41 (2s, 1H, NH). ^13^C NMR (DMSO-*d*_6_): δ = 23.08, 23.62, 28.07, 28.17, 28.31, 29.42, 33.40, 33.61, 39.60, 40.10, 43.09, 43.18, 46.41, 46.61, 50.62, 50.91, 55.01, 57.58, 57.65, 58.28, 58.46, 59.07, 59.27, 78.50, 78.56, 118.33, 118.38, 126.54, 126.64, 127.86, 127.91, 127.93, 130.75, 130.78, 131.15, 131.18, 137.15, 137.24, 138.32, 140.26, 143.24, 143.27, 143.38, 143.40, 143.46, 153.06, 153.32, 167.67, 167.75, 171.04, 171.29, 171.33, 171.96. HR-MS (ESI): *m*/*z* calcd for C_43_H_45_N_7_O_5_Na [M + Na]^+^: 762.3374, found: 762.3379. *Tert-butyl (S)-2-((2S,4S)-2-((2-oxo-2-((4-(1,2,2-triphenylvinyl)phenyl)amino)ethyl) carbamoyl) -4-(4-(((3,4,5-tris(2-(2-(2-methoxyethoxy)ethoxy)ethoxy)benzoyl)oxy)methyl)-1H-1,2,3-triazol-1-yl)pyrrolidine-1-carbonyl)pyrrolidine-1-carboxylate* (**Boc-PO(Me)G-TPE**). According to general procedure *e* from **Boc-PO(N_3_)G-TPE** (0.15 g, 0.21 mmol), Me-G1-OAc (0.12 g, 0.18 mmol), NaSAC (0.16 g, 0.08 mmol), and CuSO_4_·5H_2_O (0.05 g, 0.02 mmol) in t-BuOH/H_2_O (*v*/*v* = 1:1), compound **Boc-PO(Me)G-TPE** was afforded as a colorless product in nearly quantitative yield (0.19 g, 63%). ^1^H NMR (DMSO-*d*_6_): δ = 1.31 (t, 9H, H-Boc), 1.42–1.76 (m, 4H, CH_2_), 2.06–2.41 (m, 2H, CH_2_), 2.79–3.15 (m, 2H, CH_2_), 3.21 (d, 9H, CH_3_), 3.31–4.03 (m, 30H, CH_2_), 4.10–4.15 (m, 6H, CH_2_), 4.39–4.51 (m, 4H, CH_2_), 5.39 (d, 2H, CH_2_), 5.30–5.47 (m, 3H, CH), 7.24 (s, 2H, CH), 6.86–7.68 (m, 19H, CH), 8.43 (d, 1H, CH), 8.80, 8.97 (2t, 1H, NH), 9.34, 9.44, (2s, 1H, NH). ^13^C NMR (DMSO-*d*_6_): δ = 23.03, 23.59, 28.10, 28.17, 28.24, 29.33, 33.80, 34.01, 43.11, 43.22, 46.37, 46.60, 50.84, 51.01, 51.77, 57.32, 57.43, 57.68, 57.95, 59.08, 68.69, 69.03, 69.69, 69.85, 69.92, 69.94, 69.98, 70.05, 71.36, 72.05, 78.51, 78.59, 108.42, 118.33, 118.38, 124.14, 124.61, 124.64, 126.56, 126.62, 126.66, 127.88, 130.75, 131.20, 137.27, 138.33, 140.29, 142.22, 143.38, 152.13, 153.30, 165.11, 167.74, 171.07, 171.80. HR-MS (ESI): *m*/*z* calcd for C_74_H_96_N_7_O_19_ [M + H]^+^: 1386.6755, found: 1386.6760. *Tert-butyl (S)-2-((2S,4S)-2-((2-oxo-2-((4-(1,2,2-triphenylvinyl)phenyl)amino) ethyl) carbamoyl)-4-(4-(((3,4,5-tris(2-(2-(2-ethoxyethoxy)ethoxy)ethoxy)benzoyl)oxy)methyl)-1H-1,2,3-triazol-1-yl)pyrrolidine-1-carbonyl)pyrrolidine-1-carboxylate* (**Boc-PO(Et)G-TPE**). According to general procedure *e* from **Boc-PO(N_3_)G-TPE** (0.40 g, 0.54 mmol), Et-G1-Oac (0.37 g, 0.54 mmol), NaSAC (0.42 g, 0.22 mmol), and CuSO_4_·5H_2_O (0.13 g, 0.05 mmol) in t-BuOH/H_2_O (*v*/*v* = 1:1), compound **Boc-PO(Et)G-TPE** was afforded as a colorless product in nearly quantitative yield (0.48 g, 65%). ^1^H NMR (DMSO-*d*_6_): δ = 1.07 (m, 9H, CH_3_), 1.31 (2s, 9H, H-Boc), 1.42–1.76 (m, 4H, CH_2_), 1.96–2.40 (m, 2H, CH_2_), 3.37–4.03 (m, 36H, CH_2_), 4.11–4.14 (m, 6H, CH_2_), 4.39–4.51 (m, 4H, CH_2_), 5.39 (d, 2H, CH_2_), 5.30–5.46 (m, 3H, CH), 7.23 (s, 2H, CH), 6.86–7.75 (m, 19H, CH), 8.43 (s, 1H, CH), 8.81, 8.97 (2t, 1H, NH), 9.34, 9.44 (2s, 1H, NH). ^13^C NMR (DMSO-*d*_6_): δ = 15.18, 23.03, 23.60, 28.11, 28.17, 28.25, 29.33, 30.79, 33.81, 39.10, 39.27, 40.10, 43.11, 46.38, 46.60, 50.85, 55.01, 57.33, 57.44, 57.96, 59.08, 59.32, 65.63, 68.70, 69.02, 69.31, 69.86, 69.99, 70.05, 72.06, 78.51, 78.60, 108.41, 118.33, 118.38, 124.15, 124.61, 124.64, 126.56, 126.62, 126.66, 127.88, 127.93, 130.71, 130.78, 131.15, 137.18, 131.20, 137.27, 138.34, 140.29, 142.22, 142.29, 143.24, 143.28, 143.40, 143.47, 152.14, 153.10, 153.31, 165.11, 167.69, 167.74, 171.08, 171.80, 171.30, 173.67. HR-MS (ESI): *m*/*z* calcd for C_77_H_101_N_7_O_19_Na [M + Na]^+^: 1450.7044, found: 1450.7031. *(1-((3S,5S)-1-(L-prolyl)-5-((2-oxo-2-((4-(1,2,2-triphenylvinyl)phenyl)amino)ethyl)carbamoyl) pyrrolidin-3-yl)-1H-1,2,3-triazol-4-yl)methyl 3,4,5-tris(2-(2-(2-methoxyethoxy) ethoxy) ethoxy)benzoate* (**H-PO(Me)G-TPE**). According to general procedure *f* from **Boc-PO(Me)G-TPE** (0.16 g, 0.11 mmol) and TFA (0.41 g, 4.48 mmol) in dry DCM (5 mL), compound **H-PO(Me)G-TPE** was afforded as a colorless product in nearly quantitative yield (0.14 g, 95%). ^1^H NMR (DMSO-*d*_6_): δ = 1.76–2.02 (m, 4H, CH_2_), 2.33–2.39 (m, 2H, CH_2_), 2.92–3.14 (m, 2H, CH_2_), 3.21 (t, 9H, CH_3_), 3.39–3.91 (m, 30H, CH_2_), 4.10–4.14 (m, 6H, CH_2_), 4.23–4.67 (m, 4H, CH_2_), 5.37 (s, 2H, CH_2_), 5.28–5.40 (m, 3H, CH), 7.23 (d, 2H, CH), 6.87–7.37 (m, 19H, CH), 8.36–8.40 (m, 1H, CH), 9.38 (m, 1H, NH), 9.78 (s, 1H, NH). ^13^C NMR (DMSO-*d*_6_): δ = 25.23, 26.67, 27.89, 28.95, 29.10, 31.41, 34.38, 35.23, 39.64, 42.76, 45.69, 45.87, 51.17, 57.09, 57.93, 58.14, 58.62, 68.71, 69.04, 69.71, 69.87, 69.95, 70.00, 70.07, 71.37, 72.07, 108.42, 118.49, 124.15, 124.54, 126.58, 126.65, 127.91, 127.98, 129.76, 130.77, 131.28, 137.20, 138.27, 140.25, 140.27, 142.26, 142.29, 143.30, 143.45, 143.36, 152.14, 165.12, 167.19, 167.38, 170.44. HR-MS (ESI): *m*/*z* calcd for C_69_H_87_N_7_O_17_Na [M + Na]^+^: 1308.6051,; found: 1308.6060. *(1-((3S,5S)-1-(L-prolyl)-5-((2-oxo-2-((4-(1,2,2-triphenylvinyl)phenyl)amino) ethyl) carbamoyl)pyrrolidin-3-yl)-1H-1,2,3-triazol-4-yl)methyl 3,4,5-tris(2-(2-(2-ethoxyethoxy) ethoxy) ethoxy)benzoate* (**H-PO(Et)G-TPE**). According to general procedure, *f* from **Boc-PO(Et)G-TPE** (0.20 g, 0.14 mmol) and TFA (0.64 g, 5.60 mmol) in dry DCM (5 mL), compound **H-PO(Et)G-TPE** was afforded as a colorless product in a nearly quantitative yield (0.17 g, 95%). ^1^H NMR (DMSO-*d*_6_): δ = 1.06 (m, 9H, CH_3_), 1.78–2.02 (m, 4H, CH_2_), 2.31–2.39 (m, 2H, CH_2_), 2.92–3.17 (m, 2H, CH_2_),3.73–3.75 (m, 6H, CH_2_), 3.37–3.92 (m, 30H, CH_2_), 4.10–4.14 (m, 6H, CH_2_), 4.22–4.44 (m, 2H, CH_2_), 4.53–4.59 (m, 2H, CH_2_), 5.37 (s, 2H, CH_2_), 5.27–5.39 (m, 3H, CH), 7.23 (d, 2H, CH), 6.86–7.37 (m, 19H, CH), 8.36–8.40 (2s, 1H, CH), 8.54 (m, 1H, NH), 9.78 (s, 1H, NH). ^13^C NMR (DMSO-*d*_6_): δ = 15.20. 23.56, 27.93, 34.39, 39.10, 42.85, 45.83, 51.20, 57.11, 57.94, 58.57, 65.65, 68.72, 69.33, 70.08, 72.09, 108.42, 118.39, 126.65, 127.98, 130.78, 131.29, 137.24, 138.26, 140.27, 142.26, 143.47, 152.16, 158.15, 165.13, 167.41, 170.49. HRMS (ESI): *m*/*z* calcd for C_72_H_93_N_7_O_17_Na [M + Na]^+^: 1350.6520, found: 1350.6523. 

## Data Availability

Not applicable.

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
