# Peer review of "Dual-Responsive Supramolecular Chiral Assemblies from Amphiphilic Dendronized Tetraphenylethylenes"

_molecules, 2023, doi:10.3390/molecules28186580_

Round 1
Reviewer 1 Report
In this paper Zhang et al. reported design and synthesis of Amphiphilic Dendronized Tetraphenylethylenes.
These macromolecules can form supramolecular chiral assemblies which show pH responsive behavior. The paper is interesting although it lacks many control experiments to support all the claims made by the authors. Hence it cannot be published in molecules in its present form. The author should work on the following points before resubmitting the paper.
(i) It is not clear why from moving from Et (H-PO(Et)G-TPE) to Me (H-PO(Me)G-TPE) make such noticeable changes in physical properties like phase transition temperature, claimed by the authors. The authors should have provided the FESEM images for the self-assembled H-PO(Et)G-TPE as showed in figure 1c-d.
(ii) It is well known that TPE conjugated to OEG can form the amphiphilic self-assemblies then what is the role of the proline-tripeptide other than inducing chirality?
(iii) The report missing some important control experiments to establish their claim more prominently.
(iv) They should show the self-assemble propensity and structural characterization of the molecule without the dendrimer (example amine deprotected Boc-PO(N3)G-TPE)
(v) They need to prepare the dendrimer TPE conjugate without the tripeptide.
(vi) Finally compare the self-assemble properties of all the molecules to find out the actual interactions responsible for the self-assembled characteristic.
Author Response
(i) It is not clear why from moving from Et (H-PO(Et)G-TPE) to Me (H-PO(Me)G-TPE) make such noticeable changes in physical properties like phase transition temperature, claimed by the authors. The authors should have provided the FESEM images for the self-assembled H-PO(Et)G-TPE as showed in figure 1c-d.
Answer: As addressed in the manuscript, based on our previous reports, thermoresponsiveness of dendronized polymers carrying dendritic OEGs is dominated by the peripheral units in the dendrons, i.e., either ethoxyl or methoxyl terminals, due to the characteristic crowding effects to afford the heterogeneous dehydration feature, resulting in a completely different phase transition temperature (see for example, Angew. Chem. Int. Ed. 2010, 49, 5683-5687. Ref. 54). This heterogeneous dehydration characteristics remains true to afford distinguished phase transition temperatures for the aggregates from dendronized entities with low molar masses, just as described in one of our previous reports (ACS Nano 2021, 15, 20067-20078. Ref. 50), and even for the low molar mass of dendrons themselves (Phys. Chem. Chem. Phys. 2022, 24, 11848-11855. Ref. 57). Regarding TEM imaging of the aggregates from these dendronized TPEs, we were not able to obtain images with appropriate resolutions, most probably due to low contrast of dendritic OEGs in TEM or SEM measurements. Regarding, the low contrast of dendritic OEGs or TEGs in TEM or SEM measurements, please refer to U. Wiesner et. Al., Science 2004, 305, 1598-1601.
(ii) It is well known that TPE conjugated to OEG can form the amphiphilic self-assemblies then what is the role of the proline-tripeptide other than inducing chirality?
Answer: This is a good question. Our team has been dedicated to supramolecular chiral assembly of various entities with different (oligo)peptide motifs, aiming at understanding the role in their chiral induction, and at the same time, developing different chiral materials originated from different peptide motifs for possible applications. In this report, POG is the most abundant peptide sequence in collagens and have been proven intriguing in bio-applications, which was selected not only for the chirality induction in the supramolecular assembly, but also providing a chance to tailor the structures of the amphiphiles by attaching dendritic OEGs from different positions of the oligopeptide (in this report, to the hydroxyl group of the hydroproline unit) for exploring topological effects on the chiral assembly. Please refer to the text on page 2: “Different from the conventional linear arrangement for different segment in constructing the amphiphilic molecules, here we design to have both hydrophilic dendritic OEG and hydrophobic TPE segments “face-to-face” arranged beside POG, aiming at different tendency for interactions between these two segments, at the same time, providing less steric hindrance for the free amino group to exhibit enhanced pH responsiveness”. Actually, we also performed a series of researches on supramolecular chiral assembly of a similar system by attaching the dendritic OEGs to carboxyl terminal of the tripeptide to form the linear dendronized amphiphiles, which will be published separately.
(iii) The report missing some important control experiments to establish their claim more prominently.
Answer: We don’t understand this question from our reviewer for what it really means, but guess closely related to the question (iv) and (v). So, please refer to our answers in the following.
(iv) They should show the self-assemble propensity and structural characterization of the molecule without the dendrimer (example amine deprotected Boc-PO(N3)G-TPE)
Answer: As documented in the text, different from many nice reports on supramolecular chiral assembly in organic solvents from other groups, we are focused on supramolecular chiral assembly in aqueous phase. Furthermore, one key scientific point in this report is to develop supramolecular achiral assemblies with dual-responsiveness. Therefore, instead of investigation the deprotected amphiphilic entity for possibly (solely) pH-responsive assembly in organic solvents, we are more interested in obtaining chiral assemblies with characteristic thermoresponsiveness from aqueous phase. In addition, as described in the conclusion part, we are also interested in making use of dendritic OEGs for promising applications in asymmetric catalysis and chiral recognitions in aqueous phase, provided by the crowding effect from the dendritic OEGs (ref. 52). For these reasons, dendritic OEGs must be connected. Regarding assembly of amphiphilic peptides or amino acids in organic solvent, please refer to the Chem. Rev. 2015, 115, 7304–7397 (ref. 33).
(v) They need to prepare the dendrimer TPE conjugate without the tripeptide.
Answer: Similar as for the question (iv), without tripeptide, assembly of the dendronized TPEs should be similar to reports from Prof. Tang’s group, as cited in ref. 49.
(vi) Finally compare the self-assemble properties of all the molecules to find out the actual interactions responsible for the self-assembled characteristic.
Answer: As documented in the text, dendritic OEG, POG and TPE all acted differently in the assembly process, which can be tailored through modulating their solvation: dendritic OEGs provided hydrophilicity, and at the same time, afforded the assembly characteristic thermoresponsiveness; POG provided the chirality source, affording the assembly with varied induced chirality through interaction differently to TPE moieties or the dendritic OEGs; TPE provided hydrophobicity, and at the same time, offered a chance to follow the assembly through its aggregation-induced-emission (AIE) property. Please refer to the text in the last part before Conclusion: “Instead, the supramolecular chirality should be mainly related to interaction balance between TPE with dendritic OEG and TPE with POG moieties. POG can be wrapped easily by the dendritic OEG moieties due to the arrangement of different units on the dendronized TPEs. Since the OEG unit was connected to the side, not the end of the tripeptide, this makes it much easier for the OEG units to strongly interact with the tripeptide POG, which should have rendered the interaction between TPE and the tripeptide much weak and more sensitive to external environment, including temperature change or solution pH conditions.” In order to make this more clearly, we have revised the abstract. Please refer to our answer to the first question from Reviewer 2.

Reviewer 2 Report
I am writing to provide a review of the manuscript with Manuscript ID molecules-2581143 submitted to the Journal Molecules. The manuscript titled "Dual-Responsive Supramolecular Chiral Assemblies from Amphiphilic Dendronized Tetraphenylethylenes" is authored by Jianan Zhang, Xueting Lu, Wen Li, and Afang Zhang.
The study presented in the manuscript focuses on the supramolecular assembly of amphiphilic dendronized tetraphenylethylenes (TPEs) in aqueous solutions. The authors have connected hydrophobic TPE moieties to hydrophilic 3-fold dendritic oligoethylene glycols (OEGs) through a tripeptide proline-hydroxyproline-glycol (POG). This approach allows for the creation of supramolecular chiral assemblies with enhanced supramolecular chirality. The impact of different hydrophilic dendritic OEGs, as well as solution pH conditions, on the thermoresponsive behavior and aggregation-induced fluorescent emission (AIE) of the supramolecular aggregates is investigated.
The manuscript presents interesting findings regarding the modulation of supramolecular chirality and responsive behavior through variations in dendritic OEGs and pH conditions. The dual-responsive nature of the assemblies adds to the significance of the study, and the use of AIE for tuning fluorescence emission is a notable aspect.
However, there are a few areas that require clarification and improvement:
1. 1. The abstract provides a concise overview of the study, but it could benefit from additional details on the specific results and implications of the findings.
2. 2. The introduction should better contextualize the significance of the research and provide a clearer rationale for the study.
3. 3. The experimental methods and procedures should be elaborated further to ensure reproducibility and clarity for readers.
4. 4. Results and discussions should be more detailed, highlighting the specific observations and implications of the presented data.
One potential drawback of the manuscript is the lack of detailed discussion on the potential practical applications or implications of the findings. While the study focuses on the supramolecular assembly and responsive behavior of the dendronized TPEs, it could benefit from a clearer exploration of how these dual-responsive assemblies could be utilized or applied in real-world scenarios.
Additionally, the manuscript could provide more comprehensive comparisons or discussions of the results in the context of existing literature. Highlighting how the current findings contribute to or differ from previous research would enhance the overall impact of the study and provide a stronger foundation for its significance.
Furthermore, the manuscript could provide more insights into the limitations of the study, such as potential challenges or constraints that may arise in applying these dual-responsive assemblies in practical applications. Addressing these limitations would provide a more balanced and comprehensive perspective for readers and researchers in the field.
The conclusion should provide a comprehensive summary of the key findings and their broader implications.
Overall, while the manuscript presents valuable insights into the supramolecular assembly and responsive behavior of dendronized TPEs, addressing the aforementioned drawbacks would further strengthen the study's contribution and relevance in the field of supramolecular chemistry.
The English in the manuscript is generally well-written and clear, making it easy to understand the research and findings presented. The authors effectively convey their ideas and concepts, allowing readers to follow the study's progression and outcomes.
However, there are a few areas where minor improvements could enhance the clarity and readability of the manuscript. For instance, some sentences could be rephrased to improve the flow of ideas and prevent potential ambiguity. Additionally, ensuring consistent use of terminology and defining specialized terms would further aid readers in grasping the concepts discussed.
Overall, the manuscript's English proficiency is commendable, but a thorough proofreading and potential revisions to enhance sentence structure, terminology consistency, and clarity would contribute to an even smoother reading experience for the audience.
Author Response
We appreciate this reviewer for the nice comments and suggestions. The manuscript has been revised mostly according to this reviewer's comments.
- The abstract provides a concise overview of the study, but it could benefit from additional details on the specific results and implications of the findings.
Answer: Many thanks. We have revised the abstract accordingly: Supramolecular assembly of amphiphilic molecules in aqueous solutions to form stimuli-responsive entities are attractive for developing intelligent supramolecular materials for bioapplications. Here we report on supramolecular chiral assembly of amphiphilic dendronized tetraphenylethylenes (TPEs) in aqueous solutions. Hydrophobic TPE moieties was connected to the hydrophilic 3-fold dendritic oligoethylene glycols (OEGs) through a tripeptide proline-hydroxyproline-glycol (POG), to afford the characteristic topological structural effects from dendritic OEGs and the peptide linker. Both ethoxyl- and methoxyl- terminated dendritic OEGs are used to modulate the overall hydrophilicity of the dendronized TPEs. Their supramolecular aggregates exhibit featured thermoresponsive behavior originated from the dehydration and collapse of the dendritic OEGs, and their cloud point temperatures (Tcps) can be tailored by solution pH conditions. Furthermore, aggregation-induced fluorescent emission (AIE) from TPE moieties can be used as an indicator to follow the assembly, which can be reversibly tuned by temperature variation at different pH conditions. Supramolecular assemblies from these dendronized amphiphiles exhibit enhanced supramolecular chirality, which is dominated mainly by interaction balance between TPE with dendritic OEG and TPE with POG moieties, and can be modulated through different solvation by changing solution temperature or pH conditions. More interestingly, ethoxyl-terminated dendritic OEG provides much stronger shielding effect than its methoxyl-terminated counterpart to prevent amino group within the peptide from protonation, even in strong acidic conditions, resulting in different responsive behavior to solution temperature and pH conditions for these supramolecular aggregates.
- The introduction should better contextualize the significance of the research and provide a clearer rationale for the study.
Answer: The major concept developed in this report is the following: by combination of topological dendritic entities with AIE featured TPE moieties, helped with POG peptide, The supramolecular assembly in aqueous phase becomes possible and affords responsive aggregates with enhanced chirality from the peptide. Significant scientific contribution of this report also includes that thickness of dendritic structures provides characteristic shielding effect to different moieties and can tailor the assembly and their responsiveness. To response to our review, we have revised the introduction part accordingly.
- The experimental methods and procedures should be elaborated further to ensure reproducibility and clarity for readers.
Answer: Since supramolecular assembly can be affected by many factors due to its thermodynamic instability, all experimental data presented in this report have been repeated for several times, guaranteeing the reliability.
- Results and discussions should be more detailed, highlighting the specific observations and implications of the presented data.
Answer: Please refer to the revised text.
One potential drawback of the manuscript is the lack of detailed discussion on the potential practical applications or implications of the findings. While the study focuses on the supramolecular assembly and responsive behavior of the dendronized TPEs, it could benefit from a clearer exploration of how these dual-responsive assemblies could be utilized or applied in real-world scenarios.
Answer: This report is aimed to provide the concept for supramolecular chiral assembly to form dually responsive entities by combining dendritic OEGs, TPE and POG in one matter. As our reviewer said, it’s very important to have potential applications for such systems, to make the concept developed more valuable. In fact, we have performed applications of these supramolecular chiral assembly in asymmetric catalysis. Through modulating the temperature, the catalysts encapsulated can be recycled through thermally-mediated precipitation. In addition, through confinement of the dendritic OEGs, encapsulated peptides can be released in a controlled fashion through modulating solution temperature above or below the cloud points of the chiral aggregates. These will be published separately.
Additionally, the manuscript could provide more comprehensive comparisons or discussions of the results in the context of existing literature. Highlighting how the current findings contribute to or differ from previous research would enhance the overall impact of the study and provide a stronger foundation for its significance.
Answer: done.
Furthermore, the manuscript could provide more insights into the limitations of the study, such as potential challenges or constraints that may arise in applying these dual-responsive assemblies in practical applications. Addressing these limitations would provide a more balanced and comprehensive perspective for readers and researchers in the field.
Answer: please refer to our answer to question 2.
The conclusion should provide a comprehensive summary of the key findings and their broader implications.
Answer: Done.
Overall, while the manuscript presents valuable insights into the supramolecular assembly and responsive behavior of dendronized TPEs, addressing the aforementioned drawbacks would further strengthen the study's contribution and relevance in the field of supramolecular chemistry.
Answer: please refer to our answer to question 2.

Round 2
Reviewer 1 Report
The suggested control experiments are needed to get a comparative idea without that it is not clear why the tripeptide group is important.
Infact, the authors need to justify the role of two prolines and one glycine (as linker) units. why they have not considered one proline? It is not sufficient to justify the presence of the peptide unit by saying it is "a major component found in collagen". Actually that indicates it can also take part in self assembly.
Here, the peptide and the TPE both can participate in self assembly and it is important to look into their individual contribution and for that the control experiments are needed.
Previous reports also suggests that single amino acid can show the chiral induction. Hence the authors need to experimentally justify the utility of two prolines instead of one.
Without such clarifications the paper is not suitable for publication.
Author Response
Many thanks for the interesting criticism from the reviewer.
However, the comment on the tripeptide comes from misunderstanding of the present work. As we addressed in the text, this work aimed at investigation of the supramolecular chiral assembly of dendronized amphiphiles. Although these molecules were constructed by three segments, including dendritic OEGs, tripeptide, and TPE, it's clear that this work was not focused on the tripeptide. The functions of the tripeptide have been clearly addressed: (1) chiral source, (2) topological connection of different components, and (3) pH sensitiveness. The role of each amino acid in the peptide has gone too much beyond topic of the present work. We are interested in this peptide sequence is just because this is the most abundant in collagens, which should provide good chances for promising bio-applications, and which are on-going in my team. However, this have gone beyond the topic of the present paper. Certainly, all components in the amphiphilic molecules contribute to their assembly, as discussed in the text, otherwise, it'll be too strange. The key scientific innovation for this paper is what we are interested: stimuli-responsive chiral assembly. We hope our reviewer might understand and agree with these points. There are only a few cases published in this direction until to now, including one on Science from Lee’s group and one from our team on ACS Nano (all cited in the reference), we value this work very much, and submit this work to Molecules only because of the kind invitation from the editorial office.
Regarding control experiments to understand the roles for individual components in the assembly, we have addressed in the text and also in our previous answers. Since we can distinguish their way in participating in the assembly, the control experiments do not make sense. Specifically, as suggested by the reviewer, taking away any component for control experiments, the assembly of the resulted will make new stories, which are not our interest, and has gone beyond the topic of the present work, since the assembly will not be able to perform in water or the assembly will not be chiral, or not responsive. If the reviewer is interested the suggested researches, please refer to Liu’s reports, for example, as we replied previously.
Regarding the chiral induction from single amino acid in assembly, it’s true. However, this is not the topic of the present paper. Regarding the role of different prolines, please refer what have been addressed in the first paragraph. We hope the reviewer aware of these key scientific points.
Overall, we hope our reviewer might agree with us for these replies.
Reviewer 2 Report
Thank you for your continued support of Molecules.This manuscript presents a fascinating study on the supramolecular chiral assembly of amphiphilic dendronized tetraphenylethylenes (TPEs) in aqueous solutions. The revised manuscript effectively communicates its objective, which is to investigate the supramolecular chiral assembly of dendronized TPEs in aqueous solutions and how various factors such as dendritic oligoethylene glycols (OEGs) and pH conditions influence their behavior. The description of the experimental setup and methods used for the assembly, including the synthesis of dendronized TPEs, is clear and comprehensive. This provides a strong foundation for the research. Also, the manuscript effectively discusses the thermoresponsive behavior of the supramolecular aggregates, particularly the cloud point temperatures (Tcps) and their tunability based on solution pH conditions. This information is crucial for understanding the responsiveness of the assemblies. The revised manuscript is well-structured and logically organized, making it easy for readers to follow the research flow. Overall, the language and writing style are clear. However, make sure to proofread for any grammatical or typographical errors.
Author Response
Many thanks for the comments and suggestion. The writing has been smoothed with the manuscript revised accordingly, and all modifications were marked in yellow in the manuscript.